# Role of Peptide Associations in Enhancing the Antimicrobial Activity of Adepantins: Comparative Molecular Dynamics Simulations and Design Assessments

**DOI:** 10.3390/ijms252212009

**Published:** 2024-11-08

**Authors:** Matko Maleš, Davor Juretić, Larisa Zoranić

**Affiliations:** 1Faculty of Maritime Studies, University of Split, 21000 Split, Croatia; mmales@pfst.hr; 2Department of Physics, Faculty of Science, University of Split, 21000 Split, Croatia; davor.juretic@gmail.com

**Keywords:** antimicrobial peptides, design of peptides, molecular dynamics, associations

## Abstract

Adepantins are peptides designed to optimize antimicrobial biological activity through the choice of specific amino acid residues, resulting in helical and amphipathic structures. This paper focuses on revealing the atomistic details of the mechanism of action of Adepantins and aligning design concepts with peptide behavior through simulation results. Notably, Adepantin-1a exhibits a broad spectrum of activity against both Gram-positive and Gram-negative bacteria, while Adepantin-1 has a narrow spectrum of activity against Gram-negative bacteria. The simulation results showed that one of the main differences is the extent of aggregation. Both peptides exhibit a strong tendency to cluster due to the amphipathicity embedded during design process. However, the more potent Adepantin-1a forms smaller aggregates than Adepantin-1, confirming the idea that the optimal aggregations, not the strongest aggregations, favor activity. Additionally, we show that incorporation of the cell penetration region affects the mechanisms of action of Adepantin-1a and promotes stronger binding to anionic and neutral membranes.

## 1. Introduction 

Antimicrobial peptides (AMPs) play vital roles in the innate defense systems of diverse life forms [1,2]. Due to their pivotal function, substantial research is focused on investigating their potential as anti-infective drug candidates, especially in light of the escalating threat of bacterial drug resistance [3,4,5]. In general, the multifaceted roles of AMPs include bactericidal activity and modulation of the host’s immune and healing systems [6,7,8]. They are effective against biofilms, which cause about 80% of human infections [9,10]. Recent studies highlight their potential as anticancer [11], antiviral, and antifungal agents [12] and for other functions [13,14]. The main mechanism of action of AMPs is targeting negatively charged bacterial membranes, destabilizing them by forming pores or lesions [15,16]; thus, AMPs act by disrupting membrane-linked free-energy transduction [17]. Moreover, they can affect intracellular processes such as nucleic acid or protein synthesis, though this is less common [18,19,20].

Despite the discovery of a large number of AMPs (18,345 confirmed entries in dbAMP, https://awi.cuhk.edu.cn/dbAMP/introduction.php, accessed on 7 June 2024), a relatively small percentage have reached clinical trials (around four hundred), and around sixty have been approved by the FDA, with only a handful (seven, according to [21]) making it to the market [22]. These numbers highlight the significant challenges in discovering or designing applicable AMPs. Major issues include high toxicity to human cells, often leading to hemolysis, as well as reduced activity under physiological conditions such as protease degradation and rapid kidney clearance [23,24,25,26]. Additionally, high production costs and expensive experimental processes further complicate development.

### 1.1. Design of Antimicrobial Peptides

Computational design methods play a crucial role by providing a fast and cost-effective way to predict behavior and design antimicrobial peptides with high potential for application [27]. These methods may include site-directed mutation of natural peptides, de novo peptide design using template-based approaches [28], incorporation of specific motifs [29], and various modifications such as cyclization and capping [30,31]. The computational approaches used vary, including those based on quantitative structure–activity relationships [32,33] and molecular modeling [34,35]. Recently, machine learning techniques have gained prominence, as AMPs are well suited for these applications due to the availability of relatively large databases linking structure to function [36,37].

However, there is still a lack of studies that would contribute to the definition of general principles relating the sequence to the functional mechanisms of AMP action. Considering that AMPs exhibit high sequence variability, can adapt various structures depending on their environment, may have different mechanisms of action, and possess different functionalities, such studies are of high importance [38,39,40].

Accordingly, in this study, we focus on the designed antimicrobial peptides Adepantins (abbreviation for Automatically Designed Peptide Antibiotics) [41,42]. The impact of Adepantins in antimicrobial peptide research has been significant, in terms of both their potential [43] and the innovative design methods in which they have been used as examples [44,45,46,47] and as an inspiration in the development of online servers that predict peptide function based on sequence [48,49,50,51] (more details are provided in the Appendix A).

Adepantin-1 (Ad-1), the first of seven Adepantin peptides, demonstrated promising but narrow-spectrum activity against Gram-negative bacteria [41]. To broaden its efficacy, an analog, Adepantin-1a (Ad-1a), was developed, showing enhanced activity against Gram-positive strains [52]. Additionally, Adepantin-1a was fused with a cell-penetrating peptide (CPP) sequence, RRWFRRRRRR, creating the CPP-Ad-1a hybrid. This design aimed to achieve multifunctionality in human cells—providing antibacterial, antitumor, antiviral, antifungal, and anti-inflammatory effects while reducing hemolytic and toxic risks [19] (further details are provided in the Appendix A).

The sequences of peptides are listed in Table 1.

The comparison between the sequences of Ad-1 and Ad-1a that are listed in Table 1 shows that the GKHVG motif in Ad-1 has been replaced by **K**K**A**VG in Ad-1a, altering the N-terminal region. With this alteration, the number of small motifs at the N-terminal is increased from one (GKHVG) to two (GIKKA and AVGKA). Specifically, small residues form the [GAS]XXX[GAS] patterns, where [GAS] means that glycine, alanine, or serine can be in the first and/or last position of a small motif. Additionally, the double motif GLLKGLGES in Ad-1’s C-terminal is modified to GLLK**A**LGES in Ad-1a. The pH-sensitive His residue is removed, and the positive charges are increased at the N-terminus due to the substitution of Gly3 with Lys3 in Ad-1a.

Placing positive charges near the peptide termini is known to enhance translocation, which is supported by both experimental and simulation results [53,54]. Further, the role of small motifs in AMPs is well studied regarding their mechanism of action. For example, Fuselier and Wimley examined the influence of small motifs on translocation without causing pore leakage [55]. The involvement of small motifs in peptide dimerization or oligomerization within membranes has been confirmed [56]; it has also been shown that helix-helix interactions in the membrane environment become stronger due to the presence of small motifs and regular distribution of larger residues [57].

### 1.2. Associations of Antimicrobial Peptides

It can therefore be expected that small motif variations between Ad-1 and Ad-1a may influence peptide associations. In general, associations of AMPs can occur through self-aggregation, where the same peptides cooperate, or through synergy, where different peptides oligomerize to enhance their activity. Cooperativity and synergy were discovered by D.J. while working at the NIH in the laboratory for bioenergetics [58,59]. The synergy between peptides was confirmed by Williams et al. [60], De Waal et al. [61], Vaz Gomez et al. [62], and Westerhoff et al. [63] as functional synergism, where two peptides have a greater effect together than individually. Structural synergy has been extensively investigated for the case of PGLa and magainin heterodimer formation [64,65]. This synergy is not limited to this pair; for example, a dermaseptin mixture showed a 100-fold increase in antibiotic activity over separate peptides [66]. Producing synergistic peptides allows organisms to achieve the same effect with lower concentrations, protecting host cells while targeting pathogens [58].

However, research into the role of peptide aggregation in antimicrobial activity indicates that different types of aggregation impact activity in distinct ways. It has been reported that peptides such as Temporin-L have the highest activity if monomeric in aqueous solution but form molecular helical aggregates in the environment of Gram-positive bacteria membrane [67]. The same study also reported that aggregates on the Gram-negative bacteria membrane consist of peptides in β-structures, suggesting different mechanisms for different membranes. Aggregation of peptides on the Gram-negative membrane is reported to induce formation of toroidal pores in cases of LL-37 [68] and Protegrin-1 [69]. However, studies also report that strong aggregation in water, due to excess hydrophobicity, reduces the potency of peptides against Gram-negative bacteria compared to Gram-positive, as the highly positively charged area of the cluster is suggested to be prevented from progression beyond the anionic membrane [70]. Different assembly patterns [71] and cluster compositions [72] have also been reported to significantly influence the activity of peptides.

## 2. Results

Building upon our prior research on Adepantins [41,42,52] and a recent simulation study on Adepantin-1 [73], this investigation examines the impact of substitutions by comparing Adepantin-1 with its analog, Adepantin-1a, and assesses the influence of the CPP region when attached to Adepantin-1a [19] using molecular dynamics (MD) simulations. The results provide a detailed comparison of peptide behavior in single-peptide and multi-peptide interactions with anionic and neutral membranes. Additionally, the observed characteristics of peptide associations are analyzed in relation to the peptide designs.

The structures of the peptides, predicted by the C-QUARK structure predictor [74] (https://zhanggroup.org/C-QUARK/, accessed on 9 February 2024), and the distribution of hydrophobic and polar residues, predicted by the Heliquest program [75] (https://heliquest.ipmc.cnrs.fr/cgi-bin/ComputParams.py, accessed on 10 February 2024), are presented in Figure 1.

### 2.1. MD Simulations of Single Peptide with Anionic Membrane

Key findings from the study of Ad-1 showed that the interaction of the single peptide with the anionic bilayer (POPE:POPG membrane) follows a three-step mechanism: electrostatic binding, adaptation to the bilayer, and hydrophobic insertion [73]. Similar behavior was observed for all peptides studied. However, notable differences exist, with the process being fastest for Ad-1, followed by Ad-1a and then CPP-Ad-1, as evidenced by comparing simulation times for all three stages and as presented in Figure 2.

The trends observed by visual inspection were further investigated by mapping the time dependence of the centers of mass of clustered charged (yellow) and hydrophobic (magenta) residues. As evidenced in Figure 3 and Appendix A, in the case of Adepantin-1 and Adepantin-1a, peptides primarily bind through electrostatic interactions, with charged residues close to the membrane. Subsequently, during further interaction with the membrane, the peptides reorient, placing the hydrophobic residues closer to the membrane. Although the peptides share a similar mechanism, the time evolutions are different, as, in the case of Adepantin-1a, the simulation time for which reorientation is observed is around 2.5 ns (case1) and 1.5 ns (case2), while for Adepantin-1, it is below 1.2 ns in both case1 and case2. Additionally, DSSP results indicate that both peptides show destabilization of the initial α-helical structuring upon interaction with the anionic membrane (see Appendix A).

The plots of the density profiles (Appendix A), calculated as an average over characteristic simulation times, show the depth of membrane penetration for hydrophobic, charged residues and the entire peptide. The peptide movement toward the hydrophobic region of the membrane, facilitated by the orientation of its hydrophobic side, is visible when comparing the profiles at different simulation times (see Appendix A).

The results for CPP-Ad-1a, presented in Figure 3 and Appendix A (both on the right) indicate that for the overall peptide, a distinct pattern between charged and hydrophobic residues is less evident, more so for case1. In case2, a more separated arrangement of the charged and hydrophobic residues of the CPP-Ad-1a peptide is present during the initial binding process, but after 2.5 ns, the profiles become similar to those in case1.

The density profiles (Appendix A) show that the charged CPP region enters farther into the membrane than Ad-1 region, but the entire peptide remains at the membrane surface, partially buried in the polar membrane region for the duration of the simulation.

Furthermore, CPP-Ad-1a demonstrates distinct behavior compared to Adepantins, as the N-terminal CPP functions as a structure stabilizer, preserving the helical structure of the N-terminal region. This is evident in the DSSP plot in Appendix A; while Ad-1 and Ad-1a lose their helical structuring for CPP-Ad-1a, the helical structuring is mainly present in the CPP region bound to the membrane, and the Ad-1a region unfolds upon binding.

### 2.2. Analysis of Hydrophobic and Dipolar Moments in Single Peptide Simulations

In the literature, the amphipathic features of α-helical antimicrobial peptides are typically characterized by 2D hydrophobic moments, which serve as useful indicators, especially when the 3D structure is unknown. These measurements help differentiate how the separate polar and hydrophobic regions are.

Here, in the analysis of the 2D hydrophobic moments, as presented in Figure 1 and Appendix A, we observe almost no significant difference between Adepantin-1 and Adepantin-1a. Both peptides exhibit a nearly equal measure of amphipathic separation, with 2D-HM values of 0.596 and 0.592, respectively, for Ad-1 and Ad-1a, and hydrophobicity values of 0.275 and 0.253.

However, upon considering the 3D hydrophobic moments (3D-HM) calculated for their initial α-helical structures, it becomes evident that Ad-1a displays a higher level of amphipathicity than Ad-1. Specifically, Ad-1a exhibits a value of 19.5 ÅkT/e, while Ad-1 has a lower value of 15.6 ÅkT/e. Although direct comparison between 2D- and 3D-HM may not be straightforward, as both serve as indicators of amphipathicity, the observed differences in results suggest that subtle distinctions may not be well represented solely through 2D-HM.

Considering CPP-Ad-1a, 2D-HM may not be a suitable measure because forcing the entire peptide to be represented by a single helix might be too unrealistic as an approximation. For example, the calculated 2D-HM for the entire peptide is 0.319, which is lower than those calculated for Ad-1a (0.592) and the CPP region (0.469). On the contrary, a similar analysis for 3D-HM shows the highest value of 49.9 ÅkT/e for the whole CPP-Ad-1a peptide, 8.7 ÅkT/e for CPP, 12.8 ÅkT/e for the Ad-1a region. This is a more realistic measure of amphipathicity because the 3D-HM is highest for CPP-Ad-1a and lowest for the CPP region, which contains mostly charged amino acids and only two hydrophobic ones. Moreover, the imperfect amphipathicity of CPP-Ad1a, with tryptophan amino acid in the middle of the polar sector, decreases its 2D-HM (Figure 1), but 3D-HM has by far the [highest value for CPP-Ad1a among the three peptides. This observation is important because it is still a mystery why peptides with imperfect amphipathicity often have higher selectivity and better antimicrobial activity than their parent peptides with far better 2D-HM [53,76].

In addition, 3D-HM is a more fruitful measure considering it can be calculated as a function of time. Namely, time analyses of the 3D-HM shows that Adepantins, through adaptation and interaction with the membrane, structure in a manner which results in an increase in the value of the moment vector and places its orientation towards the membrane, suggesting a favorable position for insertion, as presented in Figure 2 (first and second columns) and in Appendix A.

We also compared 3D-HM with dipole moment (DM). The 3D-HM vector reflects the differences in the surface distribution of all nonpolar and polar regions for a given conformation of a peptide, and its orientation is towards the most nonpolar region, while DM is defined by charge distribution (for more details see Materials and Methods or reference [77,78]). The results in Figure 2 show that both 3D-HM and DM are suitable for tracking the dynamic process of peptide binding.

It is interesting to note that the values of both 3D-HM and DM exhibit a distinct pattern. The values are low during the initial binding phase, after which they increase, and subsequently, during further insertion dynamics, the values decrease but to a higher value than initially observed. As mentioned previously, these changes in the values, as well as in the orientation of the moments due to the redistribution of hydrophobic/polar regions and charges, indicate a structural adjustment of the peptide, which promotes a deeper insertion into the membrane.

### 2.3. MD Simulations of Multiple Peptides with Anionic Membrane

In simulations involving twelve peptides with anionic (POPE:POPG) membrane, a prominent feature emphasized in Adepantin-1’s mechanism is the clustering of peptides, resulting in the formation of globular structures characterized by a hydrophobic central core and a hydrophilic surface, while preserving the helical structure of the peptides [73]. These clusters, upon interaction with the membrane, dissolve and strategically position peptides with their hydrophobic sides directed toward the membrane, thereby facilitating the insertion process. Both AA-12 and CG-12 simulations of twelve Ad-1 peptides with an anionic membrane yielded the same observations, although they occurred at different simulation times.

The outcomes of simulations involving twelve Adepantin-1a peptides interacting with the anionic membrane show different behavior. The clustering of peptides is present to a lesser extent, resulting in more clusters (Figure 4, middle) that are smaller compared to those observed for Ad-1 (Figure 4, left). These smaller Ad-1a clusters also share the same organization as Ad-1 clusters, with hydrophobic residues mostly inside and polar residues mostly on the cluster surface. The same behavior is observed for the AA and CG simulations with different simulation times, where the dispersion of cluster sizes is more evident in the CG-12 Ad-1a simulations.

On the other hand, simulations of CPP-Ad-1a, performed under the same initial conditions with twelve peptides interacting with the anionic membrane, reveal chain-like and network-like clustering, as shown in Figure 4 (right).

The calculation of the cluster distribution, as well as the number of clusters and the highest number of peptides in one cluster as a function of time is presented in Figure 5 for AA simulations. In the case of Ad-1 peptides a strong preference for clustering is evident, as the cluster distribution shows the presence of small and large size clusters. Further, towards the end of the simulation, only two clusters are present, with the larger one containing ten or eight peptides in two cases of AA-12 simulations (see also Appendix A).

The preferred clusters for Ad-1a observed during simulation time are smaller, containing up to six peptides (Figure 5 (middle) and Appendix A). Evidence from the time analysis reveals mostly four stable clusters, containing at most five peptides in case1 and five clusters with mostly four peptides in case2, observed in AA-12 simulations.

The CPP-Ad-1a peptides cluster in a variety of small size clusters, up to six peptides, as shown in Figure 5 (right) and Appendix A. They do not exhibit strong clustering but rather show peptides changing connections as noted by visual inspection. This is also evident in the cluster size distribution, where clusters of different sizes are observed with similar probabilities.

In the case of the CG simulations (Figure 6 and Appendix A), faster dynamics in cluster formation are observed as expected, with the strongest clustering for Ad-1 and lesser clustering for Ad-1a. This is evidenced, for example, by the cluster size distributions where clusters of all sizes from one to ten are observed for Ad-1, while only small clusters are observed for Ad-1a (Figure 6 and Appendix A). On the other hand, the CG-12 CPP-Ad-1a cluster analysis (Figure 6 and Appendix A) confirmed that clusters are less stable than in the case of Adepantins. This is evident from the dispersion in the cluster size distribution and the time dynamics of the number of clusters and peptides in the largest cluster.

### 2.4. MD Simulations with Neutral Membrane

Simulations of the Ad-1 and Ad-1a peptides with the neutral (POPC) membrane showed low binding affinity for the neutral membrane. This is demonstrated by the center-of-mass distance calculation for AA-1 simulations, as represented in Appendix A (left and middle plots), as well as in the results of AA-12 simulations, where only a small number of peptides bind, and even then, only transiently (see Appendix A, left and middle).

The CPP-Ad-1a peptide, when interacting with the neutral membrane, shows some binding but still less than with the anionic membrane. As visible in Appendix A, in the case of a single CPP-Ad-1a peptide, the binding mostly occurs with the CPP region, while Ad-1a remains in water and only transiently binds with the C-terminal region. Similar behavior is observed in the case of twelve CPP-Ad-1a peptides interacting with the neutral membrane, where the formation of clusters that are partly bound and partly in water is observed during a simulation time of 0.5 µs (Appendix A, right).

### 2.5. Connection to Design

Key design differences between Ad-1 and Ad-1a include an increased number of small motifs and positive charges at the N-termini as well as a change in small motifs at the C-termini. When observing the characteristics of small motifs, it has been reported that there is no significant difference between glycine and other small amino acids, such as alanine and serine [57,79]. However, the findings of this study might indicate otherwise, as changes in small motifs from Ad-1 to Ad-1a seem to result in differences in interaction between the peptides upon membrane binding. This could be due to differences in oligomerization preferences among motifs such as GXXXG, GXXXA, and AXXXA. Furthermore, shifting a small motif towards the peptide terminal likely reduces its role in promoting dimerization due to the general “end-fraying” tendency of helical ends [80]. Additionally, placing positive charges at both termini, as discussed in [56], minimizes self-aggregation beyond the dimer level. Therefore, the literature findings support the view that the introduced substitutions not only strengthened the association between peptides but also reduced the number of peptides in the aggregate, highlighting the importance of changes in small-motif patterns.

The calculation of residue-pair distances presented in Figure 7 further illustrates the contribution of specific residues to peptide associations. In the representation of the distance matrices, which map the smallest distances between residue pairs, it is evident that more residues from different peptides are in contact in the case of Ad-1 (Figure 7, top row left, and Appendix A) than in the case of Ad-1a peptides (Figure 7, top row right, and Appendix A).

The total number of contacts per peptide, calculated as the average over 100 ns of simulation time and across all peptides is shown in the bottom graph of Figure 7 with individual peptide profiles displayed in Appendix A. The difference between Ad-1 and Ad-1a is apparent, while the similarity between the two simulation cases for each peptide is evident. Notably, there are more differences between the two cases of Ad-1 than Ad-1a, likely due to the greater variety in contacts when multiple peptides form a large cluster compared to when peptides form smaller clusters. Accordingly, Ad-1 peptides exhibit a higher number of contacts than Ad-1a.

The profiles of the Ad-1 sequence exhibit a noticeable increase in contacts at the C-termini, indicating a more significant role for this region in peptide clustering. While a similar trend may exist in Ad-1a, the termini in Ad-1a are less connected than the rest of the peptide, consistent with the formation of smaller clusters. When examining specific motif regions, the profiles for motifs G^3^XXXG^7^ and G^15^XXXG^19^ in Ad-1 (marked in brown) and G^1^XXXA^5^ in Ad-1a (marked in red) display a similar parabolic shape, with fewer contacts at the ending G/A residues compared to the central (X) residues within the motif. In contrast, the new motif A^5^XXXA^9^ in Ad-1a (marked in blue) displays an upward-trending profile. Additionally, Gly contacts generally show fewer connections than neighboring residues in the sequence. These differences suggest that Ad-1 may have greater contact variability along the sequence, potentially enhancing flexibility and supporting the formation of larger clusters. This analysis highlights both the similarities and the differences in the behavior of specific residues between Ad-1 and Ad-1a, shedding light on their distinct clustering behaviors.

## 3. Discussion

The main goal of the Adepantin-1a peptide’s design was to achieve a broad spectrum of activity against both Gram-positive and Gram-negative bacteria, as Adepantin-1 had only a narrow spectrum of activity against Gram-positive bacteria. The article by Juretić et al. [52] confirms that this goal was achieved and at the same time, unexpectedly a somewhat selective anti-tumor effect was observed. Namely, Ad-1, previously reported in [41] exhibited excellent activity against Gram-negative bacteria, with a MIC value of 2–4 μM against *E. coli*, but showed moderate activity against Gram-positive bacteria. In comparison, Ad-1a demonstrated stronger activity against Gram-negative bacteria, with MIC values of 1 μM and 2 μM against *E. coli*, as well as excellent activity against Gram-positive bacteria, with an MIC of 1 μM against *S. aureus* [52]. Both peptides exhibit low hemolytic activity. 

However, providing a rational explanation for the observed peptides’ biological activity and relating it to the specific substitutions that led to the design of Ad-1 and subsequently Ad-1a remains a challenge, as the very fine balance between different and/or opposite characteristics underlies peptides’ behavior.

For example, it is confirmed that the translocation of single peptides is mainly driven by the peptide’s hydrophobicity, but the process is fine-tuned by the distribution of charges. This charge distribution temporarily stabilizes a high-energy intermediate by forming salt bridges with the phosphates of the lipid headgroups [53]. In the case of Adepantins, we noticed that the implemented changes increased hydrophobicity, but this increase is observed only through 3D-HM measurements. Moreover, the Ad-1a has a net charge of +5 and Ad-1 +4, but the increase in charges did not increase the binding propensity, though it might contribute to the slower dynamics observed in simulations. When observing the single-peptide interaction with the anionic membrane, both Ad-1 and Ad-1a exhibit a similar binding mechanism, which includes initial electrostatic attraction, adaptation with an increase in 3D-HM and DM moments, orientation of the hydrophobic side toward the membrane, and deeper insertion into the polar region. In conclusion, the differences between Ad-1 and Ad-1a did not affect the single-peptide mechanism observed through MD simulations, aside from variations in time dynamics.

However, the MD simulation results revealed clear differences in peptide aggregation, suggesting that the substitutions in Ad-1a, compared to Ad-1, significantly impact peptide–peptide interactions. Namely, in simulation cases with multiple peptides, clustering is observed as a favorable characteristic, with larger globules forming for Ad-1 and smaller ones for Ad-1a peptides (see Appendix A). This difference may contribute to the distinct biological activity, which is broader and stronger for Ad-1a than for Ad-1, suggesting that the peptide’s aggregation tendency could be a key factor in the biological activity of Adepantins.

The research of the role of associations in AMP activity is still ongoing, with varied conclusions depending on the studied peptide cases. For example, according to a recent study [81], peptide self-associations exert antimicrobial activity by providing an amphipathic environment in which they can adopt a helical structure. However, Zou et al. [82] proposed that the antimicrobial activity of guanine-modified magainin II and cecropin A-melittin decreases with increased peptide self-aggregation, which contributes to the increased energy cost of embedding the peptide into the cell membrane. In our previous study with Adepantin-1, we suggested that associations might facilitate the insertion process by helping to position the peptide into a favorable conformation for insertion [73].

The findings of this study support both views. The clusters of the more active Ad-1a are smaller compared to Ad-1, and the helical structure within both Ad-1 and Ad-1a clusters is well preserved. This also highlights the subtlety required in designing AMPs: while the propensity to aggregate can be beneficial, excessive aggregation can reduce activity, as observed in the case of Adepantin. Moreover, slower dynamics and smaller cluster size for Ad-1a may lead to a higher spectrum of adaptation possibilities for Ad-1a oligomers during their interaction with different bacterial membranes.

It is also important to note that the presented results reflect only one step in the overall mechanism, with other factors likely contributing to the biological activity and potentially accounting for differences between Adepantin and its analog. Furthermore, the relationship between short time-scale and small space-scale simulation results and biological experiments is not straightforward and is often interpretative in nature [83].

Finally, this study explored the impact of adding a CPP to Ad-1a, confirming that the CPP region strengthens membrane binding and influences peptide aggregations. Specifically, in the single-peptide simulations, it was observed that the CPP region penetrates deeper into the membrane and has a stronger tendency to preserve its initial helical structure compared to the Ad-1a region, which unfolds and remains mostly on the membrane surface. Additionally, CPP binding to neutral membranes was observed, though less strongly than with anionic membranes. The addition of the CPP region also reduced the globular clustering seen in Adepantins, leading to the formation of more aligned clusters, where aggregation occurs due to the Ad-1a regions coming together in an antiparallel orientation (see Appendix A).

Overall, this study highlights aggregation as a key feature of Adepantins. Moreover, the preferred size of the observed aggregates may support cooperative and synergistic effects related to membrane permeability. The differences observed among the three peptides could influence their selectivity, paving the way for diverse applications [84].

## 4. Materials and Methods

### 4.1. Adepantin Designs and Experiments

#### 4.1.1. Details of the Design

The rationale behind deriving the Adepantins was to implement a computational method for guiding the increase in the therapeutic index (TI) of antimicrobial peptides [41]. The therapeutic index (TI) measures selectivity, calculated as the ratio of antimicrobial activity (1/MIC) to hemolytic activity (1/HC50). MIC is the concentration needed to inhibit bacterial growth, while HC50 is the concentration required to cause 50% red blood cell lysis. Therefore, a higher TI indicates a better balance between efficacy and safety.

The goal was to establish a one-parameter linear model that could correlate measured and predicted TI values for a compiled dataset of frog-derived nonhomologous AMPs. This dataset, referred to as AMPad, comprises frog-derived helical antimicrobial peptides with determined therapeutic indices defined with the MIC against *E. coli* and HC50. The construction of the linear model was strongly based on our previous work on predicting membrane-buried helices and Membrane Protein Secondary Structure Prediction Server Split 4 (available at the link http://split.djpept.com/split/4/, accessed on 5 March 2024 [85]).

The data-mining process involved identifying specific motifs, quantified by the motif regularity index, and defining the amino acid selectivity index through position-dependent rules. To account for physicochemical properties, the D-descriptor was introduced. This was calculated by converting 2D sequence profiles into vectors, bending the sequence, and performing vector summation. The D-descriptor was based on the cosine of the angle between sequence moments related to hydrophobicity scales, using a specific bending arc and weighting factor. It was useful in predicting TI improvements after point substitutions in AMPs. Prediction of the D-descriptor and TI is available on the online server http://split4.pmfst.hr/split/dserv1/, accessed on 5 March 2024. The AMP-Designer algorithm, built on 10 rules, used the D-descriptor along with statistical and experimental data such as hydrophobicity, amphipathicity, and sequence length. This methodology produced peptides Adepantin-1 to Adepantin-7 [41,42].

The Adepantin-1 analog was designed to broaden the activity of the original Adepantin-1, which targeted only Gram-negative bacteria [52]. Three substitutions—Gly3 to Lys, His5 to Ala, and Gly19 to Ala—were made to enhance selectivity and increase the hydrophobic moment, with alanine substitutions specifically chosen to boost antimicrobial activity [86]. Apart from the parent peptide, Adepantin-1a shares no similarity with other known peptides, and BLASTP search found only one match with an E-value of 5.5 or higher. The tool used was the Mutator server, which optimizes the therapeutic index (TI) of helical AMPs and can be accessed at http://split4.pmfst.hr/mutator/, accessed on 5 March 2024 [87]. There are several caveats to using the Mutator tool. First, when it predicts a very low TI, it may not suggest substitutions to improve it, requiring expert intervention for manual adjustments. Second, Mutator can be used iteratively; if initial substitutions do not yield a TI near the upper limit (95), the output can be re-entered for further suggestions. Third, if lowering MIC is the goal, expert knowledge should guide a balance between reducing MIC and achieving a high, though not maximal, TI. Finally, using Mutator for non-anuran AMPs is not recommended, as it has only been tested on anuran AMPs.

The designed CPP-Adepantin-1a [19] is a result of our efforts to identify peptides with multifunctional activity, including cell-penetrating, antibacterial, antifungal, antiviral, anticancer, and anti-inflammatory properties, along with low toxicity and hemolytic activity. Using several online servers, the specific desirable properties of peptides were analyzed (see details in Appendix A). By combining all the scores, the 20 peptides with the highest overall scores for predicted multifunctional activity and low toxicity were identified (see details in Appendix A). All of these peptides were designed by D.J. Notably, the hybrid peptide RRWFRRRRRR-Adepantin-1a ranks 19th among the top 20, underscoring its potential for further theoretical and experimental research.

Regarding the selection of the CPP region, the decapeptide was identified within the natural sequence of the hypothetical protein OLQ14316.1 from *Symbiodinium microadriaticum*, a coral dinoflagellate symbiont. The identical sequence, R(122)RWFRRRRRR(131), was also found in an uncharacterized protein (A0A5P1FK94) from asparagus (*Asparagus officinalis*). Testing the properties with online servers revealed that the RRWFRRRRRR decapeptide exhibited strong CPP activity and high uptake efficiency, suggesting its potential as a carrier for antibiotics and anticancer agents, effectively overcoming the membrane permeability barrier to target intracellular pathogens [88,89].

The results from the Mutator server show TI values of 86 for Ad-1, 95 (maximum) for Ad-1a, and 6 for CPP-Ad-1a. High antibacterial selectivity predictions for Ad-1 and Ad-1a align with experimental data, while low selectivity for CPP-Ad-1a may be due to the server’s limitations in analyzing its disordered structure [41,42,52]. Namely, this server uses the 2D hydrophobic moment, which is not suitable for disordered CPP-Ad-1a structure [19,29,90]. Therefore, the experimental verifications for predicted performance parameters for CPP-Ad-1a and other AMPs reported in [19] is needed.

#### 4.1.2. Previous Findings—Experiments

Antimicrobial activity measurements revealed that Adepantin-1 exhibited robust efficacy against Gram-negative bacteria, particularly in its C-terminally amidated form, with an MIC value of 2–4 μM against *E. coli*. Its activity against *P. aeruginosa* was moderate, with an MIC value of 16 μM. However, its effectiveness against the Gram-positive bacterium *S. aureus* was notably low, with an MIC exceeding 128 μM.

The measurements in [41] showed that even at the highest concentration tested (800 μM), the designed peptide did not induce 50% lysis of red blood cells, resulting in a therapeutic index of 200. The measurements of HC50 from 2013 reported in [42] were 480 ± 60, and the reported TI value was 190 ± 90.

It is also important to note that antimicrobial activity results may vary depending on the strain of *E. coli*. For instance, the results reported in [91] showed that Adepantin-1 was not active against *E. coli* K91BK, as well as against the Gram-positive bacteria *B. globigii* and *B. anthracis*, with MIC values greater than 100 μM in each case.

The antimicrobial activity of Adepantin-1a demonstrates excellent efficacy against *E. coli*, with MIC values of 1 μM and 2 μM, confirming its strong activity against Gram-negative bacteria, although its activity against *P. aeruginosa* was notably low, with an MIC of 64 μM. Adepantin-1a also exhibits outstanding efficacy against *S. aureus*, with an MIC of 1 μM, further confirming its potency against Gram-positive bacteria as well. The hemolytic activity was assessed by extrapolating HC50 values when all tested concentrations lysed less than 50% of red blood cells, yielding a value of 125 μM. Consequently, the therapeutic index of Adepantin-1a was estimated to be 125 [52].

To date, there have been no wet-lab experiments conducted with CPP-Ad-1a as far as we know.

### 4.2. Molecular Dynamics Methodology

Molecular dynamics simulations of antimicrobial peptides Adepantin-1, Adepantin-1a, and CPP-Adepantin-1a were carried out using Gromacs version 2021.3 (Stockholm, Sweden) [92]. Two types of simulations are made: all-atom (AA) and coarse-grained (CG) with one or twelve peptides immersed in water in vicinity of two types of membranes: anionic POPE:POPG membrane (mixing ratio 3:1, representing negatively charged bacterial inner membrane, 1-palmitoyl-2-oleoyl-sn-glycero-3-phosphoethanolamine (POPE), 1-palmitoyl-2-oleoyl-sn-glycero-3-phosphatidylglycerol (POPG)) and neutral POPC membrane (representing eukaryotic membrane, 1-palmitoyl-2-oleoyl-sn-glycero-3-phosphatidylcholine (POPC)). Simulation results for Ad-1 are presented in [73].

The POPE:POPG membrane was chosen as a simpler model for the bacterial membrane, allowing longer simulation times. This choice was also supported by the previous study that showed similar behavior for Ad-1a when simulated with a negatively charged POPE:POPG bilayer (modeling the inner membrane of Gram-negative bacteria) and with POPG:Lys-PG bilayer (modeling a Gram-positive plasma membrane model, where Lys-PG is lysylphosphatidylglycerol and CL is cardiolipin).

Detailed information about this simulation study is outlined in Table 2.

The C-QUARK structure predictor (https://zhanggroup.org/C-QUARK/, accessed on 9 February 2024 [74]) was used to obtain models for initial peptide structure predicting the α-helical structure (Figure 1). Models for lipids were taken from the CHARMM-GUI database (https://www.charmm-gui.org/, accessed on 10 December 2023.) [93]). The CHARMM36m force field [94,95] and TIP3 water model [96] were used in AA simulations, and the Martini 2.2 force-field for CG models [97]. The initial conformations for AA simulations were prepared using the CHARMM-GUI Membrane Builder [93], whereas initial conformations for CG simulations were created using the CHARMM-GUI Martini Bilayer Maker [98].

Peptide charge was defined at pH 7 considering a charged N-terminal amine but neutral amidated C-terminus. The Martini Bilayer Maker does not provide a variety of peptide terminal possibilities; therefore, we manually amidated the C-terminal by first charging both terminals and then releasing the charge on the C-terminal by swapping the Qa bead for a P5 bead [99]. A water layer of 4–10 nm thickness was added above and below the membrane. Systems were neutralized with K^+^ and Cl^−^ ions in 0.15 M concentration. The peptide(s) were initially placed in solution in plane parallel with the membrane surface and ~2 nm above it, with the hydrophilic and hydrophobic sides equally distanced from the membrane surface. The two cases (case1 and case2) for AA-1 and AA-12 simulations differed in the peptide’s position and orientation within the x–y plane, which is parallel to the membrane, while maintaining the same distance from the membrane. The two cases are used to verify if the results converge to the same behavior when starting from different initial configurations. Parameters for the equilibration and production run at a temperature of 310 K are the same as previously described in [73].

Equilibration was performed in six steps, according to the CHARMM-GUI recommendations. Isothermal–isochoric (NVT) dynamics were used for the first two steps, and NpT (constant pressure and temperature) dynamics were used for the other four steps. The temperature was fixed over the course of equilibration and production run at 310 K. During equilibration, various restraints were applied to the parts of the system: positional harmonic restraints to heavy atoms of the peptides, positional restraints on z-coordinates of lipid phosphorous atoms, and dihedral angle restraints applied on parts of lipids to prevent their unwanted structural change. These restraint forces were gradually reduced as the equilibration progressed [100].

Isothermal–isobaric (NpT) ensemble conditions were imposed using a Nose–Hoover thermostat and a Parrinello–Rahman barostat, with a 1.0 ps time constant for the temperature and 5.0 ps for pressure (compressibility equal to 4.5 × 10^−5^ bar) [101,102]. The leapfrog integrator time step was fixed at 2 fs, and the bonds were handled by the LINCS option. The particle mesh Ewald method [103] was used for calculation of electrostatic interaction with the coulombic cut-off at 1.2 nm and the van der Waals cut-off set to 1.2 nm with the force switch at 1.0 nm.

The Gromacs utilities clustsize, traj, density, and mdmat [92] were used for analysis of peptides‘ aggregation, distances of hydrophobic/charged residues from the membrane center, density profiles along the membrane normal, and contact maps and residue pair distance analysis, respectively. The charged residues include Lys and Arg, and hydrophobic residues include Ile, Val, Leu, Ala, Trp, and Phe. Secondary structure was determined with the DSSP tool [104,105]. APL@Voro program, which is based on Voronoi partitioning of the lipid surface for selected key atoms in lipid headgroups, was used to calculate membrane thickness [106].

Peptide amphipathicity was measured by the 2D-hydrophobic moment by HeliQuest [75], which uses the projection of a perfect helix on the 2D plane perpendicular to the central axis of the helix axis and Eisenberg scales to assign hydrophobicity to each amino acid residue (Figure 1).

The 3D-HM tool [77] calculates 3D hydrophobic moments vectors that reflect the differences in the surface distribution of all hydrophilic and lipophilic regions for a given conformation of a peptide. These calculations are based on the electrostatic potential on the surface of the molecule, determined using atomic point charges (defined by the force field). The distribution of polar and nonpolar regions on the surface of the molecule is then described using the difference between the absolute surface potential at each point and the average absolute surface potential. The HM vectors point towards the most nonpolar parts of the molecular surface, regardless of the sign of the individual charges or the total molecular charge, with the length of the vector indicating the magnitude of the HM.

The dipole moment vectors of peptides are calculated utilizing the Protein Dipole Moments Server (https://dipole.proteopedia.org/, accessed on 5 June 2024) [78]). Input pqr files with atom charges are obtained using Pdb2pqr utility from web server https://server.poissonboltzmann.org/pdb2pqr, accessed on 5 June 2024 [107] with CHARMM force field parameters (https://github.com/Electrostatics/pdb2pqr/blob/master/pdb2pqr/dat/CHARMM.DAT, accessed on 5 June 2024).

The VMD program was used as a visualization tool [108], Gnuplot (http://www.gnuplot.info/, accessed on 25 February 2024) was used for graphs, and GIMP (https://www.gimp.org/, accessed on 25 February 2024) was used for image editing.

## 5. Conclusions

Despite significant advances in the characterization of AMPs using methods such as machine learning, in our opinion, specific case studies are still needed. One of the reasons lies in a high variability of AMPs, in structure, sequence, mode of action, and other factors, due to which statistical analysis on a large dataset potentially obscures subtle features key to a particular mechanism of action. Therefore, both approaches are needed to speed the development of AMPs for use in medicine, the food industry, or other applications.

This study explores how the properties defined by 1D and 2D molecular descriptors in the design of Adepantins translate into the atomistic 3D behaviors observed in simulations. The main findings are as follows: (a) The substitutions in Adepentin-1 to form Adepantin-1a causes minimal changes to the interaction mechanism of the single peptide, with subtle differences in hydrophobicity and charge properties. (b) The key distinction arises in peptide–peptide interactions, where Adepantin-1 forms large aggregates and Adepantin-1a forms smaller ones, potentially explaining the stronger and broader biological activity of Adepantins-1a. We also investigated the binding mechanisms of Adepantins and Adepantin with an N-terminal CPP region. The simulations support the idea that the CPP region contributes to stronger binding to both neutral and anionic membranes. Currently, there is no experimental confirmation of the antimicrobial activity of CPP-Ad-1a. We hope this study will inspire further work toward the synthesis and testing of CPP-Ad-1a to validate its potential.

The study highlights the roles of amphipathicity and of small motifs in peptide associations and shows that favorable properties depend on fine-tuning, which is achieved through optimal, rather than maximal, desirable characteristics.

## Figures and Tables

**Figure 1 ijms-25-12009-f001:**
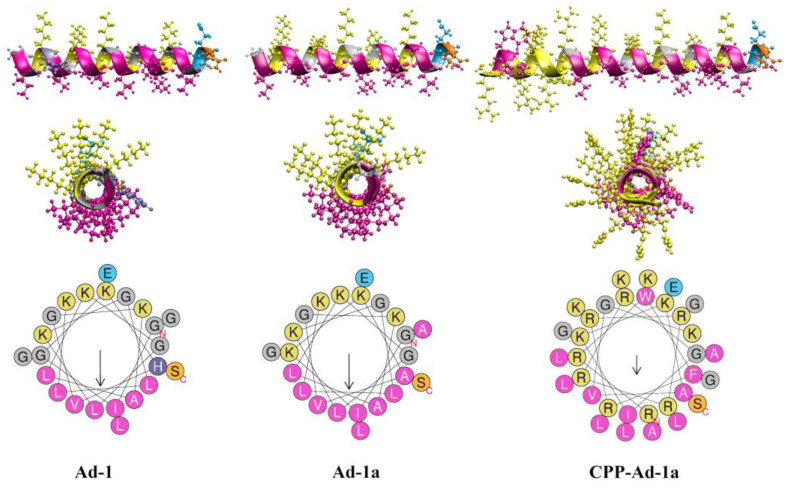
Structures of Adepantin-1 (**on the left**), Adepantin-1a **(in the middle**), and CPP-Adepantin-1a (**on the right**) as predicted by the C-QUARK structure predictor (https://zhanggroup.org/C-QUARK/, accessed on 9 February 2024). The side and top views show the peptides represented by ribbon and point models. Hydrophobic residues are shown in magenta, positively charged residues in yellow, negatively charged residues in cyan, histidine in purple, serine in orange, and glycine in gray. The bottom row presents the 2D helical wheel projections with the same residue colors (https://heliquest.ipmc.cnrs.fr/cgi-bin/ComputParams.py, accessed on 10 February 2024). More information from the Heliquest server is shown in Appendix A.

**Figure 2 ijms-25-12009-f002:**
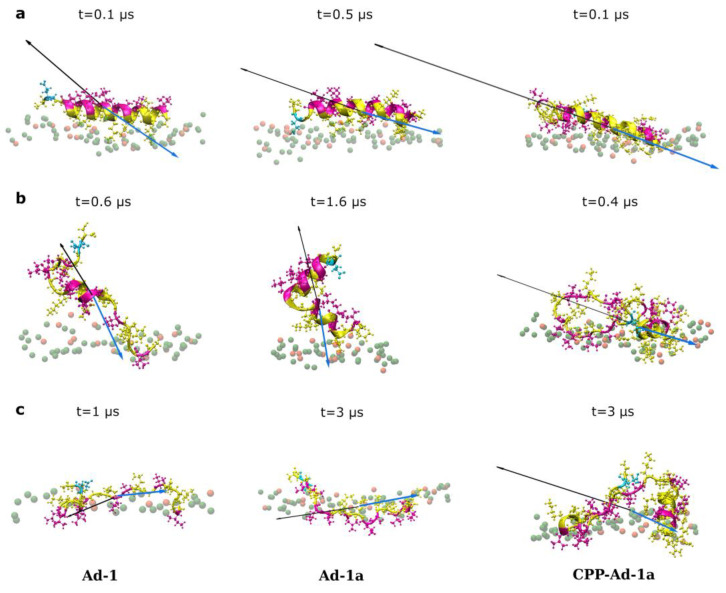
Characteristic states during peptide binding to the anionic membrane and corresponding vectors of the 3D hydrophobic moment (black) and electrostatic dipole moment vectors (blue) for Ad-1 (**left**), Ad-1a (**middle**), and CPP-Ad-1a (**right**). The first row (**a**) shows the initial electrostatic binding. The second row (**b**) shows the intermediate state for reorienting hydrophobic and hydrophilic parts. This reorientation is limited in CPP-Ad-1a due to the anchoring of charged CPP regions. The third row (**c**) shows final states, with hydrophobic parts facing the membrane in Adepantins and more complex behavior in CPP-Ad-1a. Peptides are shown in ribbon and stick–ball forms, with hydrophilic residues in yellow and hydrophobic residues in magenta. Only the upper membrane leaflet is shown, with phosphorus atoms (P) in green (POPE) and orange (POPG). Other lipid atoms and water are omitted for clarity. Vector sizes are detailed in Appendix A.

**Figure 3 ijms-25-12009-f003:**
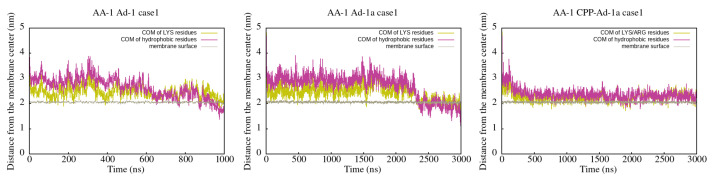
Distance between the center of mass and the membrane center for charged and hydrophobic residues for Adepantin-1 (**left**), Adepantin-1a (**middle**), and CPP-Adepantin-1a (**right**) in AA-1 simulations. More results are presented in Appendix A.

**Figure 4 ijms-25-12009-f004:**
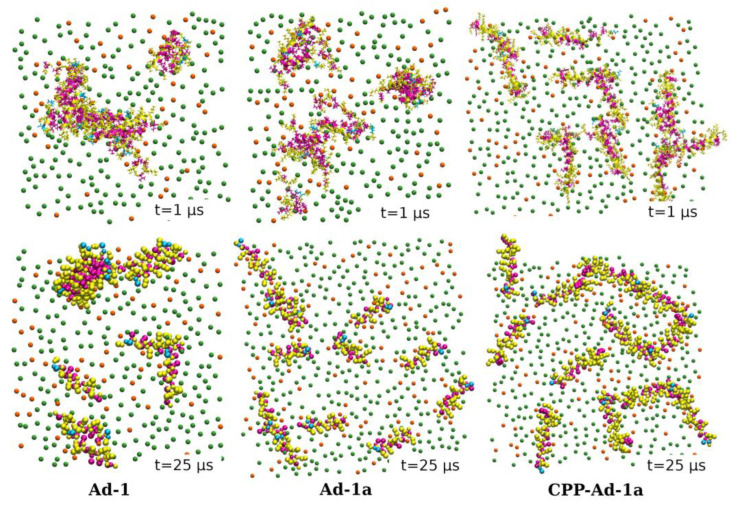
Top-view configurations and clustering of twelve peptides interacting with an anionic membrane. The simulation snapshots are taken at t = 1 μs for AA (**top row**) and t = 25 μs for CG simulations (**bottom row**). The coloring scheme for the residues is as follows: charged and polar residues are highlighted in yellow, while hydrophobic residues are shown in magenta. Small spheres represent phosphorus (P) atoms within the lipid molecules, where green spheres correspond to POPE lipids and orange spheres to POPG lipids. To enhance clarity, water molecules and other lipid atoms have been omitted from the images.

**Figure 5 ijms-25-12009-f005:**
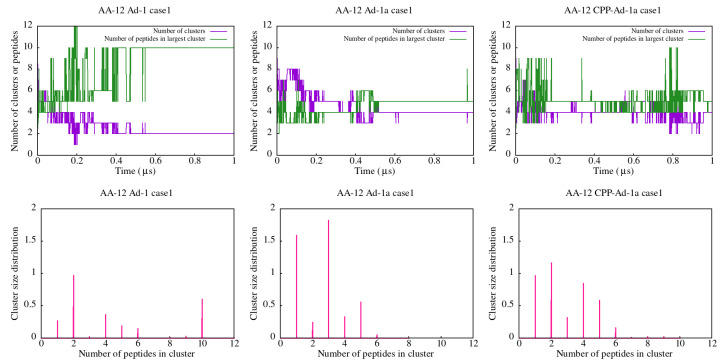
Results of cluster analysis for the AA-12 simulations of Ad-1 (**left**), Ad-1a (**middle**), and CPP-Ad-1a (**right**) peptides. The (**top row**) presents the time dependence of the number of clusters and the number of peptides in the largest cluster throughout the simulation. The (**bottom row**) displays the cluster size distributions.

**Figure 6 ijms-25-12009-f006:**
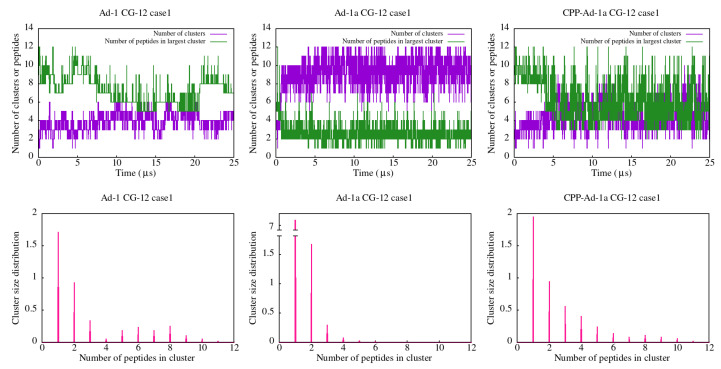
Results of cluster analysis for the CG-12 simulations of Ad-1 (**left**), Ad-1a (**middle**), and CPP-Ad-1a (**right**) peptides. The **top row** presents the time dependence of the number of clusters and the number of peptides in the largest cluster throughout the simulation. The **bottom row** displays the cluster size distributions.

**Figure 7 ijms-25-12009-f007:**
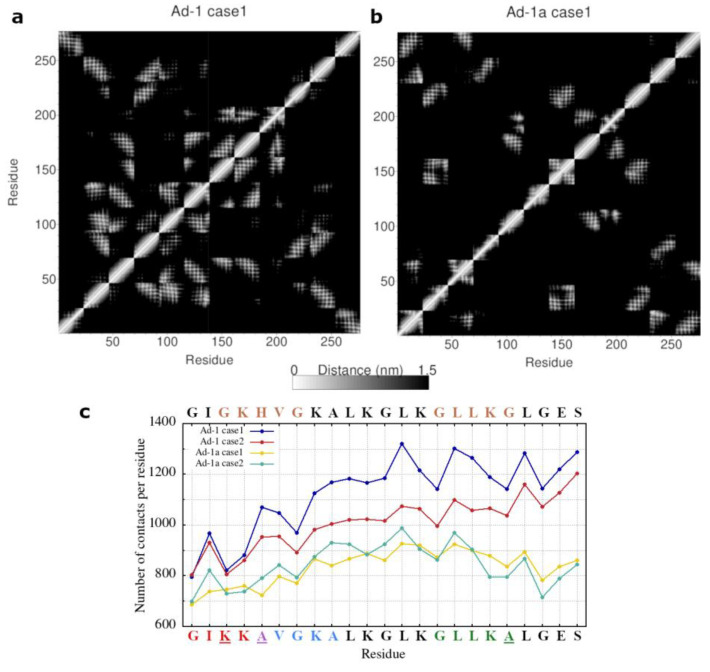
Results of the residue-pair distance analysis for the simulations of Adepantins AA-12. Parts (**a**,**b**) show the distance matrices, consisting of the smallest distances between residue pairs of all twelve peptides (a total of 23 × 12 = 276 residues), for the case1 simulations of Ad-1 (**a**) and Ad-1a (**b**). Part (**c**) presents the average number of contacts per residue for the last 100 ns of simulation time in each simulation case. Small motifs in Ad-1 sequence are marked in brown (GXXXG), while the small motifs in Ad-1a sequence are in red/green (GXXXA) and blue (AXXXA). Other results are presented in Appendix A.

**Table 1 ijms-25-12009-t001:** Peptides and their sequences. Amino acid substitutions relative to Adepantin-1 are indicated by bold, underlined letters. CPP denotes the cell-penetrating peptide addition to Ad-1a. All peptides are C-terminally amidated.

Peptide	Abbr.	Sequence
Adepantin-1	Ad-1	GIGKHVGKALKGLKGLLKGLGES–NH_2_
Adepantin-1a	Ad-1a	GI**K**K**A**VGKALKGLKGLLK**A**LGES-NH_2_
CPP-Adepantin-1a	CPP-Ad-1a	RRWFRRRRRR-GI**K**K**A**VGKALKGLKGLLK**A**LGES-NH_2_

**Table 2 ijms-25-12009-t002:** Simulations details. Duration of simulations are expressed in microseconds (μs). AA—“all-atom”, CG—“coarse-grained” simulations. Simulations with one peptide are denoted as AA-1, while AA-12 and CG-12 are simulations with 12 peptides. The two cases (case1 and case2) differed from each other in the slightly different initial positions of peptides.

Peptide	Membrane Type	AA-1 Case1 (μs)	AA-1 Case2 (μs)	AA-12 Case1 (μs)	AA-12 Case2 (μs)	CG-12 Case1 (μs)	CG-12 Case2 (μs)
Ad-1	POPE:POPG	1	1.6	1	1	25	25
	POPC	0.5	-	0.5	0.5	-	-
Ad-1a	POPE:POPG	3	3	1	1	25	25
	POPC	0.3	-	0.1	-	-	-
CPP-Ad-1a	POPE:POPG	3	3	1	1	25	25
	POPC	0.7	-	0.1	-	-	-

## Data Availability

The data presented in this study are available on request from the corresponding author.

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
