# Peer review of "Role of Peptide Associations in Enhancing the Antimicrobial Activity of Adepantins: Comparative Molecular Dynamics Simulations and Design Assessments"

_ijms, 2024, doi:10.3390/ijms252212009_

Round 1
Reviewer 1 Report (New Reviewer)
Comments and Suggestions for Authors
The manuscript present modeling and Molecular Dynamics simulations of Adepantin-based peptides. The topic is interesting and also relevant to the medical field.
The authors claim that the aggregation propensity of the different Adepantin-based peptides might explain their biological activity. Given the results of the paper, and how confusingly they are presented, I feel this explanation is not supported. Moreover, the manuscript is extremely dense with results that often seem unnecessary. Given how difficult it was to navigate the results of this manuscript, I lean towards rejection. However, I provide also my feedback and major revisions for the authors, should the editor decide to share my comments.
Why do the authors simulate the same systems in full atomistic and coarse-grained resolution?
The authors should provide experimental evidence of the Adepanting-CPP peptide.
Line 68: the authors assume the reader knows what an Adepantin-CPP hybrid is. The definition of hybrid becomes clear only at the end of the Introduction.
Table 1: Adepanin is probably a typo
Fig 1: The figure is well done. A minor issue that the authors could address is the coloring of Histidine and Glycine residues. It is very difficult to distinguish them. The choice of discussing Fig 1 in the Introduction is peculiar. Is there a reason for this?
Line 98: "Key findings": care to explain where these key findings come from?
Figure 2: how were the states defined?
Lines 122-124: The statement that 1.2 and 2.5 ns are very different numbers is highly questionable. This difference might be totally irrelevant. I see a similar irrelevance in the description of the effect of the starting configuration on the membrane insertion modality.
Lines 129-130: "accurately depict the observed behavior". What is this behavior?
Fig S4: The figure is overwhelming and almost totally unexplained in the text. Why did the authors plot the distributions at those specific times? By looking at the distributions I do not discern many differences. Most of the times, they seem very similar. I am not sure what the authors are trying to convey with these distributions. If they are so important, please perform statistical tests to ascertain that two distributions are different and (ii) explain clearly the physicochemical aspects that might cause the observed, statistically relevant only, differences.
Lines 146-160: I do not see the relevance of this paragraph and it confuses me. Are the authors claiming that a small change in the starting positions can cause a peptide to insert differently into the membrane? I am afraid that, given the very low statistics of the simulated systems in this work, the authors might be interpreting statistical variations as something physically interesting.
Comparison between atomistic and CG simulations of Adepanting 1a interacting with membrane: Fig S5b vs S5e (or Fig 5 vs Fig 6). They results do not agree at all. It's clear that the AA and CG models of Adepantin 1a give different results. Why? And also, why do the authors repeat the same figures in the SI and MS?
The Methods section has to be definitely improved. Here's why:
Line 494: It's unclear what the linear model is and references are missing to the experimental (?) data used to build the model.
Line 501: "motif regularity index" what is this index?
Line 503-504: This is another example of unclear technical descriptions. What's the definition of sequence to vector mapping? What does it even mean bending the sequence?
Line 515: It seems the author optimized the sequence to increase the TI based on a server-hosted method. Did the authors verify experimentally the claim on the antimicrobial activity? Please add proper references, if this is the case.
Line 528-531: The authors' goal sounds very ambitious and it's definitely tied to the several prediction methods the authors utilize. I would like to understand if the authors have verified that their CPP-Adepantin peptide was effective in all those therapeutic areas. If not supported by strong experimental evidence, I recommend deleting this statement and similar ones.
"Using several online servers": Can the authors be more specific? Please add a table that lists each used webserver, what the goal of each webserver was and a reference to the method. As per my previous concerns, did the authors ever verify that the predictions of each server are (i) reasonable and (ii) their "final combination" do not lead to a random result?
Line 543: "Testing the properties with online servers". Which properties and which servers?
Author Response
Please see the attachment.

Reviewer 2 Report (New Reviewer)
Comments and Suggestions for Authors
Introduction: The introduction provides a solid background on antimicrobial peptides (AMPs) and their role in combating bacterial resistance. However, it would be beneficial to elaborate on previous research concerning peptide aggregation and its implications for antimicrobial activity. Additionally, further details on the rationale for specific amino acid substitutions in Adepantin-1a (Gly3→Lys3, His5→Ala5, etc.) and how these modifications are expected to affect peptide-membrane interactions would enhance the context.
Methods: The methods are well-described, with an appropriate balance of all-atom (AA) and coarse-grained (CG) simulations. However, it would be useful to explain the choice of simulation times and the specific membrane compositions (POPE
and POPC) in more detail. Justifying the selection of the cell-penetrating peptide (CPP) region sourced from dinoflagellates and explaining how it enhances antimicrobial activity would also strengthen the methodology section.
Results: The results clearly show the differences in peptide aggregation behavior between Adepantin-1, Adepantin-1a, and CPP-Adepantin-1a. However, providing more interpretation of how these findings translate to biological relevance could make the results more impactful. For instance, explaining why smaller peptide clusters (as seen in Adepantin-1a) might lead to improved antimicrobial activity would provide readers with a more comprehensive understanding.
Conclusion: The conclusion is supported by the results, but it could be more impactful by discussing future experimental validation of the findings. In particular, testing the biological activity of CPP-Adepantin-1a could be emphasized, as it remains untested in experimental settings despite promising computational results.
Language: While the manuscript is generally well-written, moderate language editing is recommended to improve clarity in some of the technical explanations.
Figures and Visuals: The figures are informative, but additional visual aids, such as a comparison of the aggregation patterns or a diagram summarizing the key molecular dynamics steps, could help further clarify the differences in peptide behavior for readers.
Comments on the Quality of English Language
The overall quality of the English language is good, but moderate editing is recommended to improve clarity and flow in some sections. There are a few areas, particularly in the methods and results, where the sentence structure could be more concise, and certain technical descriptions can be reworded for better readability. Improving these aspects would enhance the reader’s understanding of the study's findings.
Round 2
Reviewer 2 Report (New Reviewer)
Comments and Suggestions for Authors
Thank you for attending to the concerns adequately
Comments on the Quality of English LanguageThe is good as it is
This manuscript is a resubmission of an earlier submission. The following is a list of the peer review reports and author responses from that submission.
Round 1
Reviewer 1 Report
Comments and Suggestions for Authors
Dear authors,
I am writing regarding your manuscript „Optimizing Antimicrobial Activity of Adepantins: Comparative Molecular Dynamics Simulations and Design Assessments” sent to the International Journal of Molecular Sciences, which describes MD simulation of Adepantin-1, Adepantin-1a and an adepantin/cell penetrating peptide hybrid (CPP-Ad-1a).
I have the impression that the article is not suitable for being published in Int. J. Mol. Sci. in its current form and I will describe the reasons for my opinion below:
- MD simulations of adepantin-1 have already been presented before (Membranes 2022, 12, 891). The derivative Adepantin-1a has been developed in an earlier study, showing a broader spectrum of activity. The two peptides show different behavior in analysis of clustering in the MD simulation, however a clear explanation regarding the different activity of the two peptides based on the data presented is difficult. The peptide CPP-Ad-1a has not been synthetized before and the sequence is a suggestion for further studies. In my opinion the degree of novelty is not sufficient for publication and the interpretation of data is partly speculative. I propose two ways of increasing the value of the manuscript: 1) Either increase the number of peptide sequences in MD simulation to really be able to make a suggestion for synthesis and further analysis of antimicrobial activity. And/or 2) including experimental data (CD, NMR, …) to be able to link MD simulation data to this experimental data.
- The language of the manuscript should be improved (eg. the use of articles) in case of resubmission. I recommend to seek advice from a native English speaking person for proofreading.
- Please find below some other more specific suggestions:
1) Line 38: “only a few AMPs have been evaluated for clinical use (7 approved by the FDA) -> according to literature >400 have reached the clinical trials and >60 have been approved by the FDA – 7 reached the market.
2) The text passage from line 71 to 86 needs to be optimized regarding style and phrasing.
3) Line 91 to 97: I’m not sure about the intention of this paragraph. It describes your strategy to generate the online server, but is this part of the present manuscript?
4) Line 127 to 136: This paragraph should be erased from the manuscript.
5) Material and methods: I think that the “Material and methods” needs to be reduced. -> eg. the explanation of the therapeutic index. Also the design of Adepantin and Adepantin-1a has already been described in earlier studies and does not need repetition. Most importantly, the design of CPP-Ad-1a has been described before (lit. [18]) and has been repeated in detail in the manuscript presented. This is almost a 1:1 copy of the previous article.
Overall, I think that the topic of research (making suggestions for AMP sequences based on in-silico calculations) is very important. However, the advancements presented in the manuscript do not give enough justification for publishing in its present form. As mentioned above including more experimental data and/or increasing the number of sequences analyzed might be an option for improvement, as well as the revision of language used and phrasing of the manuscript.
Best regards
Comments on the Quality of English LanguageThe language of the manuscript should be improved (eg. the use of articles) in case of resubmission. I recommend to seek advice from a native English speaking person for proofreading.
The text is too long and especially the "Materials and methods" section should be revised.
Reviewer 2 Report
Comments and Suggestions for Authors
This paper focuses on the design and functional mechanisms of Adepantins, specialized antimicrobial peptides characterized by helical and amphipathic structures achieved through targeted amino acid selection. Employing molecular dynamics simulations, the research elucidates how these structural configurations influence peptide aggregation and antimicrobial efficacy. These highlight the critical importance of meticulously adjusting amphipathic properties and peptide motifs to maximize antimicrobial performance, providing essential insights for the ongoing development of antimicrobial peptides in various applications. Several points to address are listed below.
1. The introduction contains multiple independent paragraphs. To enhance the readability and coherence of the manuscript, it is recommended to incorporate transitional phrases between these thematic paragraphs to improve the logical flow of the text. For instance, in lines 24-39, the functionality of antimicrobial peptides is introduced followed by a discussion on the importance of designing new peptides. These topics should be integrated into a single paragraph to clarify the logical connection. Additionally, many sentences lack linking words, as observed in lines 46-47, among others.
2. In lines 65-97, there is a sudden mention of your work, followed by a discussion on the importance of design methods, and then a return to the current study. Overall, the final part of the introduction seems to forcibly combine unrelated paragraphs, resulting in a disjointed text.
3. Figure 3 is referenced repeatedly in lines 175-207.
4. The results section primarily relies on MD simulations or software predictions, lacking experimental data support. Please provide corresponding experimental evidence and explain the reasons behind the observed phenomena.
5. Similarly, the discussion section features many independent paragraphs that lack logical and continuous flow. It is necessary to enhance the logical connections between paragraphs to improve the overall quality of the discourse.
6. Detailed parameter information for each step of the molecular dynamics simulations is required in lines 742-743.
Comments on the Quality of English LanguageThe English expression can be further improved.